# E-Nose and Olfactory Assessment: Teamwork or a Challenge to the Last Data? The Case of Virgin Olive Oil Stability and Shelf Life

**Margherita Modesti** [1], **Isabella Taglieri** [2] , **Alessandro Bianchi** [2] , **Alessandro Tonacci** [3,*] , **Francesco Sansone** [3], **Andrea Bellincontro** [4] , **Francesca Venturi** [2,5] and **Chiara Sanmartin** [2,5]

1. Institute of Life Sciences, Scuola Superiore Sant'Anna, Piazza Martiri della Libertà 33, 56127 Pisa, Italy; margherita.modesti@santannapisa.it
2. Department of Agriculture, Food and Environment, University of Pisa, via del Borghetto 80, 56124 Pisa, Italy; isabella.taglieri@for.unipi.it (I.T.); alessandro.bianchi@phd.unipi.it (A.B.); francesca.venturi@unipi.it (F.V.); chiara.sanmartin@unipi.it (C.S.)
3. Institute of Clinical Physiology, National Research Council of Italy (IFC-CNR), 56124 Pisa, Italy; francesco.sansone@ifc.cnr.it
4. Department for Innovation in Biological, Agro-Food and Forest Systems, University of Tuscia, Via S. Camillo de Lellis, 01100 Viterbo, Italy; bellin@unitus.it
5. Interdepartmental Research Center, Nutraceuticals and Food for Health, University of Pisa, Via del Borghetto 80, 56124 Pisa, Italy
* Correspondence: atonacci@ifc.cnr.it

**Abstract:** Electronic nose (E-nose) devices represent one of the most trailblazing innovations in current technological research, since mimicking the functioning of the biological sense of smell has always represented a fascinating challenge for technological development applied to life sciences and beyond. Sensor array tools are right now used in a plethora of applications, including, but not limited to, (bio-)medical, environmental, and food industry related. In particular, the food industry has seen a significant rise in the application of technological tools for determining the quality of edibles, progressively replacing human panelists, therefore changing the whole quality control chain in the field. To this end, the present review, conducted on PubMed, Science Direct and Web of Science, screening papers published between January 2010 and May 2021, sought to investigate the current trends in the usage of human panels and sensorized tools (E-nose and similar) in the food industry, comparing the performances between the two different approaches. In particular, the focus was mainly addressed towards the stability and shelf life assessment of olive oil, the main constituent of the renowned "Mediterranean diet", and nowadays appreciated in cuisines from all around the world. The obtained results demonstrate that, despite the satisfying performances of both approaches, the best strategy merges the potentialities of human sensory panels and technological sensor arrays, (i.e., E-nose somewhat supported by E-tongue and/or E-eye). The current investigation can be used as a reference for future guidance towards the choice between human panelists and sensorized tools, to the benefit of food manufacturers.

**Keywords:** electronic nose; food industry sensors; information technology; quality control; sensory panel

## 1. Introduction

The quality of any food product is going to worsen after a period of storage time due to various reasons, and changes in terms of safety, nutritional properties and organoleptic characteristics (appearance, odor, flavor, color, texture) are generally observed. Shelf life (SL), which could be considered a synonym of the duration of a consumer's acceptance as a function of a product's stability, differs among foodstuffs, being dependent on inherent features (formulation, microbial population, processing, packaging, etc.) and environmental factors (storage conditions such as temperature, light exposure, etc.) [1,2].

One main question is: to what extent can the modification of the quality of a product be considered acceptable? It is prohibited to sell foods that are dangerous to health, since they contain microorganisms or toxins or chemical contaminants; in addition, foods unsuitable for human consumption, i.e., with sensory and/or nutritional characteristics below the expected standard, cannot be distributed.

In this context, EU legislation (EC Regulation No. 1169/2011) established that the SL of a food, which could be defined as the "date of minimum durability", must be labeled as: i) "best before" date, related to food quality, which indicates that, after that date, there might be a decline in the organoleptic qualities of the product, but it is still safe to be consumed without risk, preferably as soon as possible, or ii) "use by" date, related to food safety, beyond which the food is no longer safe [3]. The manufacturer is responsible for defining the SL duration, also through laboratory tests, even if the legislation does not provide methods of assessment, leaving the arbitrariness of the choice to the producer.

The essential aim of an SL study is to assess the time span during which the food quality maintains an acceptable level under specified storage conditions [4], which is a crucial step for both the consumer and the producer, but its correct prediction may be difficult. The quality of a foodstuff with an overestimated SL can indeed be perceived as poor and disappointing, with consequent damage to the image of the company and economical loss. On the other hand, an underestimated SL can generate logistic problems with not only economic consequences but also ethical ones, due to the environmental and social impact deriving from an excess of food waste. According to a recent estimation, the food waste along the distribution chain spans from 25 to 50% of the worldwide food production, with huge economic, environmental but also ethical implications [5].

Several methodologies for SL assessment are available using different approaches, all based on the deep knowledge of food quality decay as a function of storage time [4]. The preliminary step of an SL study requires the individuation of the most crucial chemical, physical, or biological phenomena inducing food quality decay, in order to identify the relevant acceptability limit. When the choice of the acceptability limit is not supported by any regulatory indication, manufacturers must fix the acceptability limits as a function of their strategies and quality target [4,6].

The next step is represented by the study of the quality depletion kinetics, considering how the selected quality marker changes as a function of time during real-time SL testing (under foreseeable storage conditions) or during accelerated SL testing (storage conditions able to accelerate the degradations). The final step is the modeling of data, addressed to estimating SL [4,6].

Considering food regulation, virgin olive oil (VOO) is a unique food, whose quality categories are fixed by distinct international standards (i.e., EU legislation, Codex Alimentarius and International Olive Council (IOC)), all of whom include sensory evaluation [7].

Developed in the early nineties, the "IOC Panel test" methodology (IOOC/T.20/Doc. no. 3, 1987) for the classification of sample within a commercial category has been revised many times over the years, to improve its performance. The first aim of this sensory evaluation is to define VOOs' quality grade, to identify and quantify the main perceived off-flavors (defects), as well as their fruity scents [7].

However, sensory analysis is generally imprecise and unreproducible. Furthermore, it is time and cost consuming, it requires trained panels and it is therefore not always available in every context.

Analytical approaches, mainly represented by chromatographic (high-performance liquid chromatography, gas chromatography–mass spectrometry, gas chromatography–ion mobility spectrometry), vibrational (FTIR, MIR, NIR and Raman spectroscopy), spectroscopic (ultraviolet–visible, nuclear magnetic resonance, mass spectroscopy, fluorescence spectroscopy, $CO_2$ laser infrared optothermal spectroscopy, dielectric spectroscopy, visible spectroscopy, ultraviolet-ion mobility spectrometry) and thermal techniques (differential scanning calorimetry, thermogravimetric analyzer), provide a more precise indication of olive oil composition [8,9]. However, even in these cases some important limitations are

present: these techniques require expensive and bulky instruments, high-purity gas carriers, complex and often long sample preparation and, lastly, trained personnel, therefore limiting their use only to qualified laboratories. Additionally, the internal concentration of specific compounds does not always provide information on the resulting organoleptic sensation and, therefore, it cannot predict consumers' sensory evaluation. An alternative approach that has been proposed to overcome these drawbacks is represented by the E-nose [10–14]. The E-nose allows fast and untrained evaluation of aroma profile, very similar to that obtained from the human nose.

In this context, the present review sought to investigate the current trends regarding the usage of human panels and E-nose in the specific field of VOO SL determination, comparing the performances between the two different approaches. To better discuss the topic, a state-of-the-art analysis about the sensory SL concept, together with the mechanism of human olfactory determinations and E-nose functioning, were preliminary addressed, and a short review of the recent literature about the main applications of the E-nose in the agri-food sector was also included.

## 2. Review Methodology

Several electronic bibliographic databases (e.g., Web of Science, Science Direct and PubMed) were consulted in order to achieve the highest coverage for relevant papers published between January 2010 and June 2021. Initially, we focused our attention on the reviews, in order to critically select the main documents recently published. Then, starting from the documents selected in the predetermined time period, we included older literature sources helpful to improve and widen the topic description, reaching 346 papers. Four investigators independently evaluated the available papers, by means of predefined eligibility criteria, resolving any disagreement by discussion. The first inclusion criterion was represented by the relevance of the substance to our discussion about the state of the art, regarding the use of panel test and/or E-nose for the determination of VOO quality and SL. In the case of papers dealing with the effect of different factors (i.e., VOO quality and SL, panel test, olfactory characterization, E-nose and chemometrics, etc.), we utilized hierarchic approaches to opt for the fitting sections of discussion. A total of 147 papers were selected at the end.

## 3. Olfactory Determinations for Food Shelf Life Assessment: Principles and Main Issues

In sensory studies, human panelists work together as an instrument to measure, analyze and interpret the data collected by the five senses (sight, touch, smell, taste and hearing), and to characterize a food product during the whole production process (i.e., development of new product, quality control, consumer acceptability, flavor and taste characterization) [15]. Depending on the food analyzed, official methods for sensory analysis are defined and validated to minimize biases derived by external conditioning, such as branding and other information that can influence consumer expectation [16].

As widely reported in the literature [4,5,17,18], the quality decay of many foodstuffs, deriving from significant changes in their sensory features during storage, represents the main issue to limit SL, even before any risk to the safety of consumers occurs. Indeed, manufacturers can profitably use the information derived from sensory studies to select the best formulation or processing conditions to slow down the quality decay during storage.

For these reasons, sensory evaluations appear of utmost importance in SL determination for a lot of food categories, even in combination with instrumental or chemical analysis [5].

Among all the organoleptic parameters measured during panel tests, smell plays a fundamental role in the evaluation of food quality and SL through the perception of some specific VOCs that can be directly linked to food spoilage.

Humans and animals have all used the olfactory system to detect external risks and take countermeasures [19–21] (i.e., identifying food safety), based on personal experience and memory, as well as interpersonal characteristics, making the process strongly depen-

dent upon one's cultural background, genetics, associative learning and physiological conditions [19,22].

Actually, the underlying mechanism of smell perception appears still far from fully understood (see Bierling et al., 2021 [23] for a recent review). Indeed, smell appears as a complex phenomenon involving the intrinsic properties of a volatile, but also depending on the perceiving biological organism [24], making the investigation around it feasible through a systems approach, with the relationships between the physical space, genetic makeup of the organism, physiological activities and smell perception to be considered as a whole.

In this context, some widely accepted main points related to the olfactory perception are, until now, well defined, and can be listed below:

(i).    At least two conditions have to be addressed for a molecule to be perceived as a smell: the molecule should be volatile enough to evaporate; and the concentration of volatile compounds must exceed the threshold of perception specific for each molecule as a function of operating conditions adopted during tasting. Moreover, depending on its chemical structure, a molecule must show specific solubility to pass through the nasal mucosa (hydrophilic) as well as to bind to the olfactory receptors (hydrophobic) [23].

(ii).   Intensity is correlated positively with vapor pressure (i.e., the concentration of volatile compounds in the head space), but negatively with hydrophilicity (water solubility) [25]. Interestingly, odor discrimination seems to work independently of measuring intensity; people who cannot characterize odorant qualities as a consequence of brain lesions maintain their ability to determine odor intensity [23,26,27].

(iii).  Limited changes in the chemical structure or functional group of a volatile compound can significantly affect its smell. Thus, a model to predict odor expression from the chemical structure actually cannot be completely defined, even if odorants with the same functional group can often have similar odors [28].

(iv).   As the aroma of a mixture is different from the simple sum of its components, the whole aroma of a complex system cannot be predicted starting from the concentrations and proportions of single specific volatile compounds [29,30]. In this context, the interactions among aroma compounds in a mixture can be classified into four types: masking effect, synergistic effect, no effect and additive action [30,31]. Even if compounds showing different structures generally demonstrate masking actions, while molecules with similar aroma and structure appear to be prone to present additive action or synergistic effect, these behaviors cannot be generalized.

Starting from the above-mentioned and widely accepted evidence for the olfactory perception mechanism, it is necessary to highlight that the complete olfactory chain is complex and far from fully understood, possibly leading to several biases, as discussed in the following section:

(i).    Final odor perception be can directly influenced by a lot of external variables other than the chemical composition of molecules, that can affect the relationship between smell and chemical structure; during sniffing, the odor concentration that effectively reaches the nostrils can be affected by nasal flow characteristics [28,32,33]; odor perception can also be affected by pre-receptor events, such as the bond to odorant-binding proteins or the enzymatic conversion of odorants in the nasal mucus (e.g., conversion to acids and alcohols of aldehydes and esters) [28,34,35]. Further, one main issue in the prediction of the final smell expression of a specific olfactory stimulus is that the human olfactory perception is part of a multisensory integration among all the sensorial and social information gathered by our environment, and not a linear analytical process of molecule detection [23].

(ii).   Odorants can pass through the nasal passages (so-called orthonasal stimulation) and via the mouth (retronasal stimulation) [24]; nasopharyngeal or nasal mucus differ in composition; thus, aroma perception can significantly differ in orthonasal and retronasal olfaction, because of the different solubility of volatile compounds in the two media [36]. The perception of taste appears affected by odors over the retronasal

pathway and vice versa; thus, gustatory and olfactory experiences are generally blended [24,37].

(iii). As the number of odorous molecules that humans are able to distinguish appear to be dramatically higher than the number of olfactory receptors identified till now (from 400,000 to 1 million estimated possible odorant molecules [38] vs. only 396 unique olfactory receptors [39]), the most promising theory to represent the method of odor identification is that a small number of olfactory receptors respond to a great number of odorants in a combinatorial way. According to this view, the receptors can be broadly tuned and respond to many different odorants, being most responsive to structurally similar odorants, or narrowly tuned, responding to a small group of odorants [28,40].

(iv). The polymorphism of olfactory receptors represents the molecular basis of the extreme variability widely detected at genetical and physiological levels in the human olfaction perception for both specific sensitivity and general olfactory acuity, with sensitivity varying by several orders of magnitude between individuals [28,41].

(v). The difficulty called the "tip-of-the-nose phenomenon" [24] suggests that olfaction is often an unconscious process [32], during which humans are able to recognize smells, but they often have problems labeling them linguistically. This is likely the reason why defining smell based on olfactory perception is not intuitive and needs the recall to a visual or tactile aspect [42,43].

(vi). The close anatomic relationships between the systems deployed for olfaction and for emotion [44] account for the important links found between these two functions [45–49]. More than any other sensory modality, olfaction is like emotion in attributing positive (appetitive) or negative (aversive) valence to the environment. To objectively and quantitatively assess the physiological response to olfactory stimulation, a reasonable solution, merging acceptability, affordability and reliability, and providing useful information about the physiological reactions to odorous stimuli, is represented by the assessment of biomedical signals triggered by the activity of the autonomic nervous system (ANS), including electrocardiogram (ECG) and galvanic skin response (GSR), already studied in relation to the olfactory assessment [49–51]. Such signals can be acquired in a completely non-invasive manner using wearable sensors, as demonstrated in several works published to date [49–51]. Those signals, captured non-invasively via lightweight, affordable devices, can be particularly useful to objectively estimate the degree of emotional response to sensory stimuli in an individual.

## 4. E-Nose: Principles and Main Agri-Food Applications

In the past (up to the 1990s), sensory analysis was generally used to detect the different olfactory traits of many agri-food products, but it was often imprecise, not reproducible and too subjective. Hence, human senses, including the sense of smell, as discussed above, are influenced by physical and mental status and also by different exogenous factors. Therefore, sensory analysis was often coupled and compared with analytical instruments (e.g., gas chromatography). However, the latter requires time, expertise and expensive machinery [52]. Furthermore, this kind of analytical approach often requires sample preparation, which is complex and time consuming, and it is not always compatible with the modern food industry, which requires an easy and rapid detection tool for food quality evaluation. To overcome the above-mentioned drawbacks, in recent years, tools somewhat mimicking the biological sense of smell, namely the E-nose, have become some of the most used instruments in the food industry.

Commonly, thanks to the technological capabilities of the sensing part (the sensor array constituting the device), the E-nose is somewhat able to transform volatile compounds contained within a biological matrix or in the environment into some detectable—often digital—electronic signal, which is, in turn, properly analyzed, mostly during post-processing, to extrapolate a possible pattern of some significance for the given analysis. Under such

premises, it becomes clear that the main part constituting an E-nose tool, and that can be customized upon the desired application, is represented by the sensor array.

Historically, the majority of E-nose systems relied on conducting polymers (CP), metal oxide semiconductors (MOS), metal oxide semiconductor field-effect transistors (MOSFET) and mass-sensitive (such as quartz microbalance), acoustic and optic sensors.

Among them, MOS sensors are probably the most widely used since the early E-nose applications, due to their affordability, good performances, low tendency to drift and good sensitivity to various volatiles. Their principle of operation relies on the change in conductivity brought by the reactions between the detected volatile and adsorbed oxygen. However, they are poorly selective and quite prone to being poisoned by weak acids. On the other hand, MOSFET sensors are based on the variations in the electrostatic potential. They are considered to be particularly robust to environmental conditions, although a fine control of surrounding temperature is desirable for data interpretation. Commonly, mass-sensitive devices, such as piezoelectric sensors, take advantage of the piezoelectricity phenomenon to translate a mechanical variation due to the mass of the ligands into a change in resonance frequency. They are particularly selective, even if their stability to changes in temperature and humidity is quite poor. Finally, when it comes to CP sensors, they are considered particularly sensitive and resistant to the poisoning effects, different from MOS devices, but also feature a limited reproducibility.

Therefore, in order to maximize the positive characteristics of all those principles of operation, reducing their associated drawbacks, thanks to the technological advances in the field, mainly in terms of device miniaturization, many hybrid solutions were adopted for a plethora of applications, even in the food industry, and mainly depending on the biological matrices to be investigated. This approach goes also far beyond the E-nose principles, as it is also applied to the synergy between E-nose and E-tongue or, in some instances, E-eye, or even to the combination between E-nose and analytical instruments to provide for a more accurate characterization demanded by a given task [53].

The other pivotal functionality of the E-nose tools is represented by the data processing and interpretation, which is normally performed under the Machine Learning principles. Normally, clustering actions are carried out, by using both supervised and unsupervised approaches, depending on the specific tasks demanded. Under this light, methods such as Support Vector Machines (SVM) or Artificial Neural Networks (ANN), if not more complex algorithms such as the ones applied for Deep Learning (DL), are to be considered among the most popular when it comes to use within E-nose systems.

E-nose holds many advantages with respect to traditional sensory analysis: it is more accurate and reliable, it is less time consuming and it is less influenced by environmental factors. Furthermore, if compared with analytical methods, it does not require strong expertise to be used and interpreted as gas chromatography does. As such, the application of the E-nose in various fields, including the food and pharmaceutical industries, as well as in health and well-being, is expanding rapidly [54].

As the olfactory fingerprint provides important information regarding the characteristics, origin and processing methods of food, the E-nose systems have been widely studied and used in the agri-food sector (Table 1).

**Table 1.** Main applications of the electronic nose in the agro-food sector.

| Category | Application | Sensor Arrays | Chemometrics Approach | Classical Methods for Comparison | Reference |
|---|---|---|---|---|---|
| **Discrimination of variety and ripening stage** | Discrimination of two varieties of galangal (*Alpinia officinarum*) | MOS | PCA | GC-MS | [55] |
| | Dogfruit (*Pithecellobium jiringa*) and stink bean (*Parkia speciosa*) ripening stages | Chemical sensors | PCA, HCA | GC-FID | [56] |
| | Discrimination of three varieties of garlic (*Allium sativum* L.) | MOS | PCA | GC-MS | [57] |
| | Discrimination between mango (cv Chokanan) ripening stages | MS based (piezoelectric quartz crystal) | PCA, HCA | GC-FID | [58] |
| | Identification of five *Piper nigrum* L. genotypes | MOS | PCA | HS-SPME GC-MS Sensory analysis | [59] |
| | Identification of three mango varieties (*Manguifera indica* L.) and ripening stage | MOS | DFS | GC | [60] |
| | Discrimination of two tomato (*Lycopersicum esculentum*) ripening stages (i.e., green stages and ripe) | MOS | PCA, LDA and DFA | Fruit quality characteristics such as: soluble solids content, pH and maximum puncture force | [61] |
| | Discrimination of eight varieties of apricot (*Prunus armeniaca*) | MOS | PCA and FDA | LLE-SPME GC-MS Sensory analysis | [62] |
| **Freshness evaluation, flavor and aroma** | Monitoring volatile constituents of cocoa (*Theobroma cacao*) during the refining process | MOS | PCA | HS GC-MS | [63] |
| | Evaluation of freshness of broccoli during storage (*Brassica oleracea* L.) | MOS | PCA, CDA | HS GC-MS, FTIR | [64] |
| | Characterization of volatile compounds in soybean seeds (*Glycine max* L.) | MOS | PCA | HS-SPME GC-MS | [65] |
| | Monitoring the hardness of litchi under different storage conditions | MOS | LDA, CCA, BPNN-PLSR | Physicochemical index parameters (i.e., soluble solids content, titratable acidity and pH value) | [66] |
| | Evaluation of banana maturity | MS based (piezoelectric quartz crystal) | PCA, MLR | Respiratory quotient, total soluble solids, firmness and moisture content | [67] |
| | Monitoring of pineapple (*Ananas comosus*) shelf life during storage at different temperatures | MOS | PCA, CA | | [68] |
| | Aroma development during ripening and storage of apricots | MOS | PCA | Firmness, total soluble solids, pH, GC-MS and sensory analyses | [69] |
| | Evaluation of maturity and shelf life of tomatoes (*Lycopersicum esculentum*) | MOS | PCA, LDA, PLS | Firmness | [70] |
| | Apple and orange post-harvest quality, e.g., detection of defects of apples and oranges | MOS | PCA, PLS-DA | Amount of mealiness and skin damage | [71] |
| | Classification of 90 different blended and roasted coffee samples | MOS | DFA, MANOVA | | [72] |

**Table 1.** *Cont.*

| Category | Application | Sensor Arrays | Chemometrics Approach | Classical Methods for Comparison | Reference |
|---|---|---|---|---|---|
| **Spoilage evaluation** | Identification of early infestation of *Bactrocera dorsalis* in citrus (*Citrus reticulate*) | MOS | PCA, LDA | | [73] |
| | Detection of pathogen (*Salmonella, Erwinia, Streptococcus* and *Staphylococcus*) contamination of apples | MOS | PCA, HCA | HS-GC-MS | [74] |
| | Decay detection of peach (*Prunuspersica* L. *Batsch*) during storage | MOS | PLSR, LS-SVM, MFRG | Visual evaluation of rot | [75] |
| | Detection of diseased blueberry fruit inoculated with grey mold (*Botrytis cinerea*), anthracnose (*Colletotrichum gloeosporioides*) and Alternaria rot (*Alternaria* sp.) | CP | PCA | | [76] |
| | Identification of spoiled tomatoes (inoculated with *Aspergillus and Penicillium* spp.) | MOS | PCA | DHS-GC-MS | [77] |
| | Classification of damaged and infested apples (*Malus domestica*) | CP and MS based (piezoelectric quartz crystal) | CMAES, PCA, PNN | | [78] |
| **Wine evaluation** | Early detection of smoke taint in wine grapes, while not perceivable by sensory evaluation | MS based (quartz microbalance) | PCA | | [79] |
| | Discrimination between fermented and unfermented musts | MS based (quartz microbalance) | PCA | GC-MS | [80] |
| | Comparison of threshold detection performance and concentration quantification with a trained human sensory panel | MOS | PCA | Sensory analysis | [81] |
| | Monitoring postharvest controlled partial dehydration | MS based (quartz microbalance) | PCA | GC-MS | [82] |
| | Calibration transfer applied to the analysis of wine aroma using synthetic wine prepared from the most common wine aromas | MS based | PLS | | [83] |
| | Classification of Tempranillo wines according to geographic origin | MS based | PCA, PLA, SLDA | | [84] |
| | Identification of geographical origin of Sauvignon Blanc wines, with GC-MS that was then used to train an LDA | MOS and MS based | LDA | GC-MS | [85] |
| | Discrimination of beer and wines tainted with off-flavors | MOS | PCA, DFA | | [86] |
| | Five different wines elaborated with the major varieties from the DO vinos de Madrid were used for testing the discrimination capability of the developed system | MOS | PCA | GC-MS | [87] |
| | Characterization of different wine fruits (blackberry, cherry, raspberry, blackcurrant, elderberry, cranberry, apple and peach) based on their odor profiles | MOS | PCA and DFA | GC-FID | [88] |
| | Monitoring of aroma production during wine must fermentation | CP | PCA | HPLC, GC-MS | [89] |

E-nose finds application in different steps of the agri-food production chain, such as process monitoring, harvesting time evaluation, storage condition and SL evaluation. The latter includes the assessment of freshness or decay degree, microbial contamination and off-flavor formation [90]. The odor constituents of fruits vary greatly during ripening and storage (mainly in terms of terpenes, alcohols and esters); therefore, the aromatic composition at a specific moment can provide important information about the maturity level and can be efficiently discriminated through E-nose tools [58,60,61]. An important shift in the VOCs profile, indeed, can occur with different storage or processing protocols, such as roasting applied to coffee or cocoa [63,72] or partial controlled dehydration of grapes [82]. Additionally, food pathogens produce specific off-flavors from their metabolism and often lead to significant decay of the organoleptic features of the products, resulting in economic loss, consumer rejection and risks for consumers' health. Many studies have proved that such VOCs can be studied to assess for an early bacterial and fungi contamination of fresh or minimally processed agri-food products during storage [63]. As for every approach discussed herein, the need for a rapid, reliable and affordable method for early screening of contaminated agri-food is nowadays particularly important, not just to protect consumers' health, but also to avoid economic losses. In this context, the capability of some E-nose tools to discriminate among infected and healthy fruits and vegetables is often declared to reach classification rates up to 99.9% [64,70]. Behind that, defects such as over-ripening, damage on product surface and biochemical and physiological post-harvest mechanisms can lead to changes in aroma. E-nose is reliable and sensitive enough to correctly predict different common changes naturally occurring in post-harvest [64,91,92].

E-nose tools can possibly be replaced, in some instances, by specific sensors for the quantitative assessment of the presence of a given volatile compound or a small group of them. Indeed, in some cases, the degradation of food quality can be controlled through the production of one, or few waste products, as occurring, for example, in the case of ripened pork salami, where flavor deterioration is associated with abnormal levels of 2-heptenal and methyl esters of heptanoic, pentanoic and hexanoic acids [93]. Another common example is that related to nuts, where the rancid flavor associated with the deterioration of walnut oils can be associated with the production of 2-octenal, hexanal, 2-heptenal, 1-octen-3-ol, hexanoic acid and nonanal, different from the almond oils, where lipid oxidation is more related to 1-pentanol, hexanal and hexanoic acid, and from the peanuts, whose degradation is marked by octanal, nonanal, hexanal and 2-pentylpyridine [94].

## 5. Case Study: Panel Test and E-Nose for the Determination of VOOs' Shelf Life

### 5.1. EVOO Quality and Main Stability Issues

The term virgin olive oil (VOO) shall be taken to mean "the oil extracted from the fruit of the olive tree (*Olea europaea*), solely by means of physical or mechanical process".

Therefore, in order to produce a VOO, olives must be subjected only to washing, decanting, centrifugation, and filtration; oils produced by chemical extraction, re-esterification or oil blending processes do not fall within this definition [7,95]. VOO quality, in terms of chemical features and sensory profiles, is affected by several elements, such as genetic features, pedoclimatic conditions, training system, harvest period, extraction procedure and storage conditions [96–101].

It is therefore possible to classify VOOs into three main categories according to their chemical and sensory parameters (Table 2): (i) extra virgin olive oil (EVOO), recognized as the highest grade of olive oil with beneficial health effects, (ii) VOO and (iii) lampante olive oil (LOO), which is destined for technical uses or refining processes only.

**Table 2.** Quality standards of virgin olive oil as requested by current legislation (EU Reg. (EEC) 2568/91 as amended, that also establish the official methods for their determination [7]).

| | Free Acidity | Peroxide Index | $K_{232}$ | $K_{270}$ | $\Delta K$ | Fruity Median | Defects' Median |
|---|---|---|---|---|---|---|---|
| | % Oleic Acid | mEq O/kg | | | | | |
| EVOO | $\leq 0.8$ | $\leq 20$ | $\leq 2.50$ | $\leq 0.22$ | $\leq 0.01$ | $> 0$ | $= 0$ |
| VOO | $\leq 2.0$ | $\leq 20$ | $\leq 2.60$ | $\leq 0.25$ | $\leq 0.01$ | $> 0$ | $< 3.5$ |
| LOO | $> 2.0$ | - | - | - | - | - | $> 3.5$ |

A VOO is composed essentially of triglycerides (98–99%, distributed as follows: monounsaturated fatty acids (MUFAs = 65–83%), polyunsaturated fatty acids (PUFAs = n − 6: 6–15%; n − 3: 0.2–1.5%) and saturated fatty acids (SFAs = 8–14%)), with minor constituents such as phenols (i.e., phenolic alcohols, such as hydroxytyrosol and tyrosol; secoiridoids such as oleocanthal and oleuropein derivatives; lignans such as pinoresinol), tocopherols, terpenic acids, sterols, 4-methylsterols, carotenoids, chlorophylls, mono and diglycerides, free fatty acids, esters and volatiles, which give peculiar sensory properties and several bioactive functions [102–104]. Water is also present in very small, but essential amounts as micro-droplets, which are associated with polar or amphipathic substances belonging to the minor constituent's group [102].

The volatile molecules constituted by five or six carbon atoms, produced in the *lipoxygenase* pathway, represent the biggest part of volatile compounds in VOOs and exert an essential role in the expression of the green attributes of VOO [105,106].

Among these, worthy of particular attention for their desirable effects, are hexanal (cut grass), guaiacol (soapy, olive paste), octanal (citrus, lemon), (E)-2-decenal (soapy, fatty), 1-penten-3-ol (green plants, grassy), (Z)- 3-hexenyl acetate (fruity), (E)-2-hexenal (green) and 6-methyl-5-hepten-2-one (nutty) [95]. Nevertheless, there are also several compounds responsible for negative organoleptic perception and belonging to the class of "off-flavor" (i.e., rancid, winey, fusty, vinegary, frozen), generally caused by the oxidation process [95].

Moreover, due to the high content of MUFAs and PUFAs, along with the presence of enzymes such as *lipase* and *peroxidase*, VOO lipids are susceptible to oxidation and enzymatic hydrolysis, which promote autoxidation [107]. This degradative process reduces VOO's nutritional and healthy effects, together with its economic value, since it is the main cause of VOOs downgrading to a lower quality classification (i.e., EVOO in VOO or LOO).

Many factors influence the lipid oxidation process during production and storage: artificial light exposure, sunlight, storage temperature, humidity and air exposition, as well as packaging features, together with the oil chemical composition [97,98,108–113].

As reported in Figure 1, VOO compounds (i.e., fatty acids or liposoluble vitamins) are oxidized and turned into unstable products which ultimately trigger further degradation reactions, forming off-flavors and toxic compounds [6,114].

In this context, VOO SL could be defined as the storage time during which safety, sensory and quality parameters are within accepted limits for the specific commercial category (see Table 2) [107].

Since VOOs produced in one crop season are usually consumed before the next crop season [115,116], it is necessary to minimize oil deterioration during the storage period. The assurance of the stability of EVOOs is a matter of great concern for the olive industry [115]. In this context, it is mandatory for the olive oil industry to monitor oil quality throughout the production line and to be able to provide realistic information about the stability and the SL.

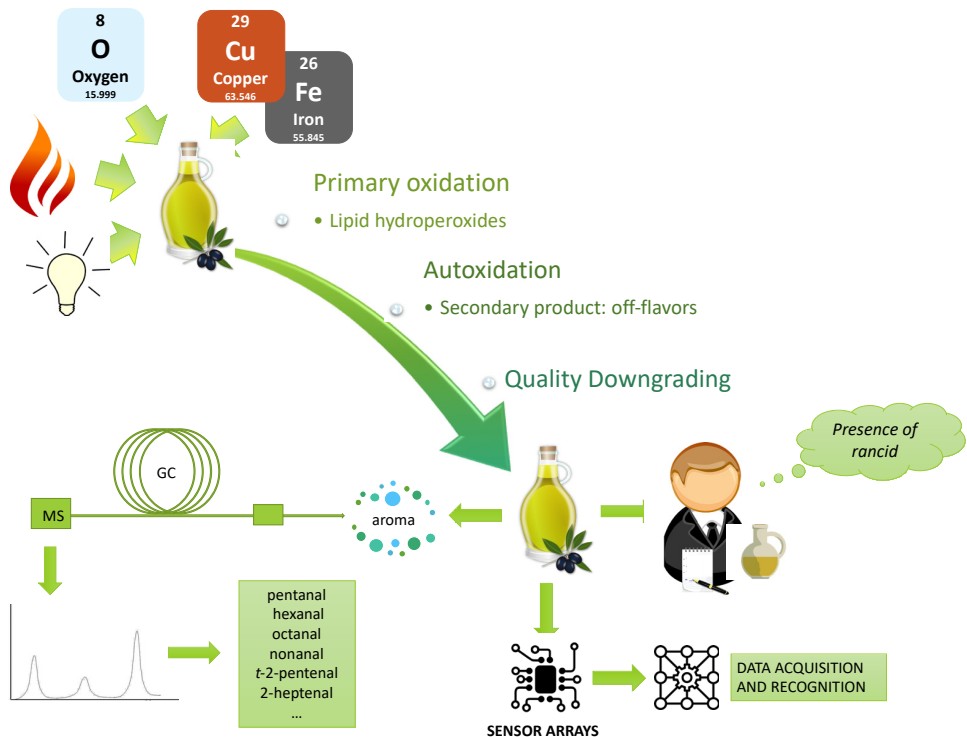

**Figure 1.** The exposition of VOOs to oxygen or catalyst agents induces the oxidation of unsaturated fatty acids, with the formation of unstable hydroperoxides, which, during autoxidation, form off-flavors (secondary volatile products such as aldehydes, ketones, alkenes and alkanes) responsible for the VOO quality decay.

### 5.2. Virgin Olive Oil Shelf Life Assessment

Nowadays, different methods are available to monitor the quality decay of VOOs as a function of oxidation process; for many of them, the current legislation defines the acceptability limits based on instrumental and/or sensory methods. Besides the conventional parameters (Table 2) in the literature, many examples of stability and SL studies are available, both in accelerated or real-time conditions, based on the VOO aromatic traits as well as on the loss of bioactive compounds [117] (Table 3).

**Table 3.** Non-compulsory indices which can be assumed as markers to track the evolution of the oxidative degradation of virgin olive oils during storage and relevant analytical methodologies [6,96].

| Quality Indicator | Method |
| --- | --- |
| Tocopherols | HPLC |
| Polyphenols | COI/T.20/Doc No. 29 |
| Oleuropein aglycon content | HPLC |
| Hydroxytyrosol content | UNI 11702:2018 |
| Tyrosol content | UNI 11702:2018 |
| Carotenoids | Spectrophotometer |
| Degradation products of chlorophyll a | ISO 29841:2009 |
| Vocs analysis | HS-SPME-GC/MS; sensor arrays |
| Hexanal | HS-SPME-GC/MS |
| In vitro antioxidant activity | Spectrophotometry |
| Color | Spectrophotometry Image analysis Color analysis |

According to the main focus of this review, only papers dealing with sensory analysis and sensor arrays for the VOO SL assessment will be taken into account.

According to EU legislation, the panel test still represents the only official method (EEC 702/2007) to detect the increase in off-flavor in VOO for the quality classification of VOO samples, based on the organoleptic detection or not of the fruity attribute, together with the indication of the median value of the predominant defect [118].

Starting from this point, it seems extremely important that the accredited laboratories perform panel tests of VOO in compliance with a well-established quality system by applying the laboratory organization and technical conditions of analysis requirements (COI/T.20/Doc. No. 15/Rev. 8, 2015), together with the guidelines fixed in the Norm ISO 17,025 (COI/T.28/Doc. No 1, 2007—revised in 2017) [7].

As described in these guidelines, an accredited laboratory is asked to correctly monitor the panel proficiency and to periodically check the validity of its entire quality system, in terms of reproducibility and repeatability of the tests, to individuate and solve possible systematic errors early. With this aim, certified reference materials are available to support laboratories for both the correct training of panelists as well as to provide a useful tool to give a measure of the accuracy evaluations carried out during the panel test.

Genovese and co-workers [105] recently published an interesting review about VOO flavor, with particular emphasis on the mechanisms affecting its production and release during tasting. Interestingly, the authors highlighted that the differences in VOO organoleptic characterization during storage, as reported by different panels, can be explained by the interaction among positive (i.e., bitter, pungent, fruity, green) and negative (i.e., rancid, fusty, moldy, winery) attributes during VOO tasting. For example, the high concentration of phenolic compounds that generally characterize fresh VOOs can completely mask the "fusty" character of the sample. In this case, the "fusty" defect cannot be detectable by panelists until the phenolic concentration decreases because of oil oxidation during storage.

Although the panel test still represents the most effective and complete tool for the quality classification of a VOO [119,120] during production and storage, the correct application of VOO sensory assessment appears time consuming and difficult to apply during routine operations, especially considering small- and medium-sized enterprises [121,122].

Consequently, the development of instrumental methods for rapid screening could support the sensory analysis in discriminating samples close to the quality border (EVOO/VO and VO/LO). To this end, several promising analytical instrumental techniques, especially based on the detection of VOC markers, have been developed [123], and many E-nose tools with different chemical sensors, system designs and methods of data analysis have been tested for olive oil analysis.

In this context, owing to its quickness, the flash gas-chromatography E-nose in combination with chemometrics is being revealed as one of the most encouraging and useful screening methods to support the sensory analysis [123].

E-nose equipped with QCM sensors was used to discriminate between different VOO categories (i.e., edible and non-edible olive oils) [124]. MOS and piezoelectric sensors were also used to discriminate over 140 samples of olive oil (VOO, non-virgin and seed oil) [125,126], whereas CP sensors were used to distinguish between olive oils of different qualities. This technique has been found to provide a good degree of selectivity and also the identification of specific features, considered as a quality fingerprint [127]. Eight compounds were identified by using an E-nose (i.e., 4-methyl-2-pentanol, (E)-2-hexenal, 1-tridecene, hexyl acetate, (Z)-3-hexenyl acetate, (E)-2-heptenal, nonanal and a-farnesene) to study the aroma fingerprint of 15 different EVOOs [128].

As far as SL evaluation is concerned, the degree of oxidation of volatiles is a key point for the use of E-nose. As discussed above, storage conditions (especially temperature and light) play a key role in maintaining olive oil quality over time. Many studies reported that it is indeed possible to discriminate between oils stored in different conditions. An E-nose based on 32 CP sensors was used to determine the rancidity of EVOO caused by the auto-oxidation process during storage [129]. The results obtained from the measurements

of different Italian olive oils by an E-nose with EC sensors show that oils can be easily distinguished between them. Additionally, changes in the aromatic profile due to the handling of the sample itself can be monitored. This aspect is of particular interest, considering the possibility to monitor changes during the ageing of the olive oil and whether it has been properly stored [130]. In a different study, a commercially available E-nose (equipped with MOS-FET sensors) was used to assess the oxidation degree of olive oil samples: EVOO was stored at 40 °C in the dark for two months and volatile compounds (by E-nose) and peroxide value (for comparison) were evaluated. The E-nose was able to discriminate among the different storage conditions and the results correlate well with peroxide values [11,131]. Furthermore, other authors [132] used the E-nose equipped with 10 MOS sensors to distinguish oil samples characterized by different degrees of rancidity and fruitiness. The results obtained agree with those found during the intensity rating test carried out by a traditional panel test. Furthermore, the study demonstrated the capability of the E-nose in the monitoring of the evolution of oil flavor during storage. Marchal et al. (2021) [133] utilized a commercial E-nose (10 MOS sensors) to predict the intensity of the fruity attribute and off-flavors in VOOs, proposing to apply it for a fast screening of VOO quality.

The authors of [134] compared the degree of oxidation in olive oil stored for up to 2 years in various lighting conditions. The results obtained using an E-nose equipped with MOSFET sensors are as precise as those obtained using reference methods (Reg EU 2568/1991). Lerma-García et al. (2010) [131] were able to detect aromatic defects in olive oil (namely fusty, moldy, muddy, rancid and winey) and their value with MOS sensors. The sensory analysis performed by trained panelists was shown to be more robust if associated with the E-nose. The authors of [135] demonstrated that, thanks to a combination of an E-nose, an E-tongue and an E-eye, the analysis of olive oil bitterness can be successfully performed. As such, many applications of E-nose have been tested for olive oil authenticity assessment and SL monitoring in recent years, and are summarized in Table 4. Of course, further research is still needed to develop advanced sensors with superior capabilities and sensitivity for olive-oil-specific traits; thus, making the evaluation faster and more accurate [117]. This topic is increasingly earning attention, since there is the necessity to develop and refine the existing analytical techniques to make them more reliable, fast and inexpensive. In particular, it is crucial to guarantee high quality traits over time, accessible to all the players of the olive oil production chain. In this sense, the applications of the E-nose, such as those discussed above, represent a promising opportunity to overcome the many limitations which still characterize the traditional techniques.

**Table 4.** Main applications of the electronic nose in the olive oil discrimination and evaluation.

| Category | Application | Sensor Arrays | Chemometrics Approach | Quantitative Classification Performances | Classical Methods for Comparison | Reference |
|---|---|---|---|---|---|---|
| Discrimination | Discrimination of edible and non-edible VOO | QCM | PCA | 99% | Acidity, peroxide value, content of oxidation compounds and panel test | [123] |
| | Classification of vegetable oils | MOS | LDA | 95.8–100% | | [124] |
| | Discrimination between VOO, non-virgin and seed oil | MOS and MS based (piezoelectric quartz crystal) | PCA, RBF | 95–99% | | [125] |
| | Discrimination of quality, variety of olive and geographic origin | CP | PCA | 96.3% of variance explained by the first 3 PCs | Acidity, peroxide value, content of oxidation compounds and panel test | [126] |

**Table 4.** *Cont.*

| Category | Application | Sensor Arrays | Chemometrics Approach | Quantitative Classification Performances | Classical Methods for Comparison | Reference |
|---|---|---|---|---|---|---|
| **Flavor evaluation** | Identification of different aromatic fingerprints | MOS | PLS-DA, PCA | 84.6–99.5% | GC-MS | [127] |
| | Analysis of olive oil bitterness | MOX | PCA | Around 0.9 correlation between electronic methods and sensory panels | HPLC and panel test | [134] |
| **Shelf life evaluation** | Evaluation of EVOO rancidity and oxidation | CP | PCA, MDA | Up to 98.6% of variance explained | Panel test | [128] |
| | Evaluation of aromatic changes during ageing and storage | EC | PCA | N/A | | [129] |
| | Monitoring of OO oxidation during storage | MOS and MOSFET | PCA | >90% | Peroxide value | [11] |
| | Evaluation of oxidation degree of VOO stored in different conditions | MOSFET | LDA, ANN | >99.1% | Peroxide value and panel test | [130] |
| | Evaluation of rancidity in VOO and monitoring of bottled VOO shelf life | Chemical sensors | PCA | 78–78.7% of variance explained | GC-MS | [131] |
| | Evaluation of different degree of rancidity and fruity flavor | MOS | PCA | 88% | Acidity, peroxide value, content and panel test | [132] |
| | Evaluation of flavor evolution during storage | MOSFET | LDA | 100% | Acidity, peroxide value, content of oxidation compounds and panel test | [133] |

## 6. Conclusions

Among the legal standards for identity (chemical composition) and quality (free acidity, peroxide value, UV absorbency and sensory evaluation) discrimination, sensory evaluation is one of the most important methods to differentiate high-quality from low-quality olive oil [7]. Sensory assessment, however, needs many resources and time, as well as specialized panelists, which are not always available to small/medium-sized enterprises and cooperative companies, and should not be used for routine operations [121,122].

In this context, there is a need for the development of accurate instrumental techniques able to perform real-time measurements and generate the same information as a panel, in a reproducible and stable way, aiming to rapidly and efficiently achieve the correct VOOs classification [136]. The development of easy-to-use E-noses, designed for directly equipping the process line, appears extremely appealing in this field. E-nose tools are more and more used in the food industry concerning tasks related to the SL assessment of various edible products, and are partially replacing human panelists in such characterizations.

However, the main limitation for the use of the E-nose in the agri-food sector concerns sample preparation and sampling methods. Some of the E-nose sensors are extremely sensitive to environmental conditions (such as temperature, humidity, pressure and vapor). These factors also greatly affect the amount of volatiles released by the samples; therefore, preparation and method of sampling can deeply affect the output [137]. For this

reason, albeit being controllable even during the post-processing and signal analysis phase, accurately controlled conditions are preferable for samplings, and therefore, it is quite difficult, or imprecise, to use the E-nose in the outdoor setting [63]. Another limitation for such an approach is that, to have a reliable result, a good sample set (typically not less than 10) is generally required. Additionally, other samplings (with a sample set often even greater than 10) are required to train and validate the classification algorithms [63]. Other steps, which can represent a limitation, include post-acquisition analysis, such as PCA and HCA, which require time and expertise. The comparison between the data reported in the literature represents another challenge. Hence, the great differences in sensor type, sample type and preparation, sampling methods, recognition algorithm and training system often result in unreproducible approaches. Therefore, the direction for future E-nose approaches is mainly to minimize sample preparation methods and optimize sampling stability conditions. As such, there is a requirement for new sensing materials less sensitive to environmental conditions, but with great affinity to volatile compounds. Another possible study direction is represented by the creation of an online library to store data from all users, to make the E-nose's application more universal, repeatable and user-friendly [63].

In conclusion, especially when it comes to the punctual assessment of the quality of olive oil, one of the main pillars of the well-known "Mediterranean diet", a synergistic approach, merging the advantages of objective characterization enabled by the E-nose tools (and possibly also the use of E-Tongues/E-Eyes) with those derived from the huge experience and ultra-fine sensory profiles of human panelists, appears to produce the best results. With fast improvements in the technological fields of material sciences (concerning the sensing parts) and in the Machine/Deep Learning algorithms (concerning data analysis), the E-nose-based tools will likely soon be able to completely replace human beings in this complex task, assuring superior performance, higher reliability and, ultimately, cost savings.

**Author Contributions:** Conceptualization, F.V., C.S. and A.T.; Writing—original draft writing M.M., I.T., A.B. (Alessandro Bianchi) and F.S.; review and editing, F.V., C.S., A.B. (Andrea Bellincontro), F.S. and A.T.; supervision, F.V. and C.S.; funding acquisition, F.V. and A.B. (Andrea Bellincontro). All authors have read and agreed to the published version of the manuscript.

**Funding:** This research was funded by Ministero Dell'istruzione, Dell'università E Della Ricerca (Progetto Violoc FISR2019_03020); "Departments of Excellence-2018" Program (Dipartimenti di Eccellenza) of the Italian Ministry of Education, University and Research, DIBAF Department of University of Tuscia, Project "Landscape 4.0—food, wellbeing and environment".

**Institutional Review Board Statement:** Not applicable.

**Informed Consent Statement:** Not applicable.

**Data Availability Statement:** Not applicable.

**Conflicts of Interest:** The authors declare no conflict of interest.

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
