# Peer review of "E-Nose and Olfactory Assessment: Teamwork or a Challenge to the Last Data? The Case of Virgin Olive Oil Stability and Shelf Life"

_applsci, doi:10.3390/app11188453_

Round 1

Reviewer 1 Report

In this paper, Modesti et al. present an overview of e-nose approaches in food science, especially to detect spoilage and to asses shelf life. They present a special case study of virgin olive oils and compare the assessment of e-nose and of expert panel. The topic is interesting, relevant for the food science community, and presents a good overview of recent applications of e-noses. I suggest the paper is published when my comments are addressed. Mostly, I believe the paper should be revised language- and formatting-wise since there are several awkward sentences and incorrect formatting cases. Further comments below.

  • The first sentence is strange, »Electronic Nose (E-nose) devices are representing one of the most trailblazing innovations in current technological research, since mimicking the functioning of the biological sense of smell still remains a gap technological advances are yet to resolve.« Innovations, and yet to resolve. I suggest a rewrite.
  • Abstract, line 5, has strange word spacing.
  • In particular, the food industry has seen the pervasive advent of such technological tools for determining the quality of edibles, progressively replacing human panelists,… This is also a strange sentence.
  • »sensors« is a very vague keyword, as are »panel« and »ICT«
  • The quality of any food product is going to get worse to fail … Awkward formulation
  • In general, foods which are dangerous for health, since they contain microorganisms or toxins or chemical contaminants, as well as foods unsuitable for human consumption, i.e. with sensory and/or nutritional characteristics below the expected standard, cannot be sold. This is again an awkward formulation, since there are 3 lines of text between the beginning and the end point. This could easily be written as: It is prohibited to sell foods that are dangerous to health, because of …
  • In studies of VOO and EVOOs, several experimental techniques have been used to study the quality, adulteration, aging, and similar. You just mention gas chromatography, but there have been several recent papers dealing with UV, Raman, NMR, rheology, and similar. I suggest a paragraph is added in the introduction to present a brief overview of other techniques, apart from e-noses. It would fit this review paper well.
  • In Review methodology, I am missing some numbers about how many papers were screened and how many papers were selected for the inclusion in this review.
  • During the evolution, organisms used the olfactory system as an alert mechanism to interpret signaling from the environmental context … It took me a while to get the point of this sentence, which is the evolution of olfaction in living beings. This sounds rather out-of-scope for a technical paper. The following discussion about the perception and things that influence the sense of smell looks a bit too long for a paper about e-nose for VOOs. The list of main points later on looks ok, though.
  • Page 10, here, in place would be a short discussion regarding the dilemma between several sensors and one specific sensor for a particular compound with a very high sensitivity. For several food-related applications, it is a common approach to track just a single compound that is indicative of, for example, spoilage.
  • 5: Probably “Case study”, not “Case of study”
  • It would be nice to list the VOO papers in the table in the same manner than for other food applications, to have them easily searchable.
  • I am missing some more discussion about the efficiency of e-noses in VOOs in the papers mentioned. Did they use PCA or machine-learning? What was the performance?

Author Response

We are extremely grateful to the Reviewer for their constructive comments, allowing us to improve the quality of the manuscript, overall.

Here, a point-to-point response to the reviewers is present, displaying their comments (in plain text), our response (in italics) and the parts added in the manuscript (undersigned).

Reviewer 1

In this paper, Modesti et al. present an overview of e-nose approaches in food science, especially to detect spoilage and to asses shelf life. They present a special case study of virgin olive oils and compare the assessment of e-nose and of expert panel. The topic is interesting, relevant for the food science community, and presents a good overview of recent applications of e-noses. I suggest the paper is published when my comments are addressed. Mostly, I believe the paper should be revised language- and formatting-wise since there are several awkward sentences and incorrect formatting cases. Further comments below.

Thank You for Your kind observations. We took all of them into account and revised the paper accordingly (see the point-to-point response below and the part highlighted in red throughout the manuscript). The revision of English language/grammar and typos was performed by a native speaker.

  • The first sentence is strange, »Electronic Nose (E-nose) devices are representing one of the most trailblazing innovations in current technological research, since mimicking the functioning of the biological sense of smell still remains a gap technological advances are yet to resolve.« Innovations, and yet to resolve. I suggest a rewrite.

Thank You. We agree, and rephrased it as “Electronic Nose (E-nose) devices are representing one of the most trailblazing innovations in current technological research, since mimicking the functioning of the biological sense of smell has always represented a fascinating challenge for technological development applied to life sciences and beyond.”

  • Abstract, line 5, has strange word spacing.

Thank You. Corrected.

  • In particular, the food industry has seen the pervasive advent of such technological tools for determining the quality of edibles, progressively replacing human panelists,… This is also a strange sentence.

Thank You. We modified as: “In particular, the food industry has seen a significant rise in the application of technological tools for determining the quality of edibles […]”

  • »sensors« is a very vague keyword, as are »panel« and »ICT«

The keywords proposed in the original version of the manuscript were representing “major” categories. Under your suggestion, we modified them making them more specific. This was done by deleting “food industry” and replacing “sensors” with “food industry sensors”. “ICT” was changed into “Information Technology”, whereas “panel” was replaced by “sensory panel”.

  • The quality of any food product is going to get worse to fail … Awkward formulation

Thank you. We modified into “The quality of any food product is going to worsen […]”

  • In general, foods which are dangerous for health, since they contain microorganisms or toxins or chemical contaminants, as well as foods unsuitable for human consumption, i.e. with sensory and/or nutritional characteristics below the expected standard, cannot be sold. This is again an awkward formulation, since there are 3 lines of text between the beginning and the end point. This could easily be written as: It is prohibited to sell foods that are dangerous to health, because of …

Thank You. We modified as “It is prohibited to sell foods that are dangerous to health, since they contain microorganisms or toxins or chemical contaminants; in addition, foods unsuitable for human consumption, i.e. with sensory and/or nutritional characteristics below the expected standard, cannot be distributed.”

  • In studies of VOO and EVOOs, several experimental techniques have been used to study the quality, adulteration, aging, and similar. You just mention gas chromatography, but there have been several recent papers dealing with UV, Raman, NMR, rheology, and similar. I suggest a paragraph is added in the introduction to present a brief overview of other techniques, apart from e-noses. It would fit this review paper well.

Thank You. According to the suggestion, the introduction was modified adding the lacking information.

  • In Review methodology, I am missing some numbers about how many papers were screened and how many papers were selected for the inclusion in this review.

Thank You, we agree with the Referee and added the missing number.

  • During the evolution, organisms used the olfactory system as an alert mechanism to interpret signaling from the environmental context … It took me a while to get the point of this sentence, which is the evolution of olfaction in living beings. This sounds rather out-of-scope for a technical paper. The following discussion about the perception and things that influence the sense of smell looks a bit too long for a paper about e-nose for VOOs. The list of main points later on looks ok, though.

Thank You for Your observation. Your concern was also ours, when we drafted the paper, since we would have liked to remain focused on the main topic of the manuscript, at the same time providing useful guidance to the reader about the reasons why panels are so important also in nowadays’ practice and which are their main drawbacks technology could help outdoing. However, based on your observation, we reduced this part whenever possible, in order to avoid distraction to the reader from the main points of the article.

  • Page 10, here, in place would be a short discussion regarding the dilemma between several sensors and one specific sensor for a particular compound with a very high sensitivity. For several food-related applications, it is a common approach to track just a single compound that is indicative of, for example, spoilage.

Thank You for Your extremely useful observation. We added some example to discuss this point at the end of the paragraph 4. This part now reads: “Possibly, E-nose tools can be replaced, in some instances, by specific sensors for the quantitative assessment of the presence of a given volatile compound or a small group of them. Indeed, in some cases, the degradation of the food quality can be controlled through the production of one, or few waste product as occurring, for example, in the case of the ripened pork salami, where flavor deterioration is associated with abnormal levels of 2-heptenal and methyl esters of heptanoic, pentanoic and hexanoic acids [Lorenzo et al., 2013]. Another common example is that related to nuts, where the rancid flavor associated with the deterioration of walnut oils can be associated with the production of 2-octenal, hexanal, 2-heptenal, 1-octen-3-ol, hexanoic acid and nonanal, differently from the almond oils, where lipid oxidation is more related to 1-pentanol, hexanal, and hexanoic acid, and from the peanuts, whose degradation is marked by octanal, nonanal, hexanal and 2-pentylpyridine [Valdés Garcìa et al., 2021].”

  • 5: Probably “Case study”, not “Case of study”

You are right, thank You. Changed.

  • It would be nice to list the VOO papers in the table in the same manner than for other food applications, to have them easily searchable.

We agree with Reviewer, thank You. Done.

  • I am missing some more discussion about the efficiency of e-noses in VOOs in the papers mentioned. Did they use PCA or machine-learning? What was the performance?

Thank You. Such information were all added in the newly included Table 4.

Reviewer 2 Report

The manuscript discuss the limitation for the use of the E-nose and and olfactory for the stability and shelf life of virgin olive oil in agri-food sector. It is well organized and will have a significant contribution to the field. There are some typo mistakes. The English language and style are fine/minor spell check required.  For example: ...VOCs marker have been developed[126] and many E-nose tools with chemical different chemical sensors, system design, and data analysis have been tested for olive oil analysis.

Author Response

We are extremely grateful to the Reviewer for their constructive comments, allowing us to improve the quality of the manuscript, overall.

Here, a point-to-point response to the reviewers is present, displaying their comments (in plain text), our response (in italics) and the parts added in the manuscript (undersigned).

Reviewer 2

The manuscript discuss the limitation for the use of the E-nose and olfactory for the stability and shelf life of virgin olive oil in agri-food sector. It is well organized and will have a significant contribution to the field. There are some typo mistakes. The English language and style are fine/minor spell check required.  For example: ...VOCs marker have been developed[126] and many E-nose tools with chemical different chemical sensors, system design, and data analysis have been tested for olive oil analysis.

Thank You. We double-checked and corrected the typos and the language throughout the manuscript through the support of a native speaker.

Round 2

Reviewer 1 Report

The paper has been improved in the revision process. I have no further issues and I support the publication.